# Unsupervised Few-shot Learning via Self-supervised Training

## Abstract

Learning from limited exemplars (few-shot learning) is a fundamental, unsolved problem that has been laboriously explored in the machine learning community. However, current few-shot learners are mostly supervised and rely heavily on a large amount of labeled examples. Unsupervised learning is a more natural procedure for cognitive mammals and has produced promising results in many machine learning tasks. In the current study, we develop a method to learn an unsupervised few-shot learner via self-supervised training (UFLST), which can effectively generalize to novel but related classes. The proposed model consists of two alternate processes, progressive clustering and episodic training. The former generates pseudo-labeled training examples for constructing episodic tasks; and the later trains the few-shot learner using the generated episodic tasks which further optimizes the feature representations of data. The two processes facilitate with each other, and eventually produce a high quality few-shot learner. Using the benchmark dataset Omniglot, we show that our model outperforms other unsupervised few-shot learning methods to a large extend and approaches to the performances of supervised methods. Using the benchmark dataset Market1501, we further demonstrate the feasibility of our model to a real-world application on person re-identification.

## 1 Introduction

Few-shot learning, which aims to accomplish a learning task by using very few training examples, is receiving increasing attention in the machine learning community. The challenge of few-shot learning lies on that traditional techniques such as fine-tuning would normally incur overfitting (Wang et al., 2018). To overcome this difficulty, a *set-to-set* meta-learning(episodic learning) paradigm was proposed (Vinyals et al., 2016). In such a paradigm, the conventional mini-batch training is replaced by the episodic training, in term of that a batch of episodic tasks, each of which having the same setting as the testing environment, are presented to the learning model; and in each episodic task, the model learns to predict the classes of unlabeled points (the query set) using very few labeled examples (the support set). By this, the learning model acquires the transferable knowledge (optimized feature representations) across tasks, and due to the consistency between the training and testing environments, the model is able to generalize to novel but related tasks. Although this set-to-set few-shot learning paradigm has made great progress, in its current supervised form, it requires a large number of labeled examples for constructing episodic tasks, which is often infeasible or too expensive in practice. So, can we build up a few-shot learner in the paradigm of episodic training using only unlabeled data?

It is well-known that humans have the remarkable ability to learn a concept when given only several exposures to its instances, for example, young children can effortlessly learn and generalize the concept of "giraffe" after seeing a few pictures of giraffes. While the specifics of the human learning process are complex (trial-based, perpetual, multi-sourced, and simultaneous for multiple tasks) and yet to be solved, previous works agree that its nature is progressive and unsupervised in many cases (Dupoux, 2018). Given a set of unlabeled items, humans are able to organize them into different clusters by comparing one with another. The comparing or associating process follows a *coarse-to-fine* manner. At the beginning of learning, humans tend to group items based on fuzzy-rough knowledge such as color, shape or size. Subsequently, humans build up associations between items using more fine-grained knowledge, i.e., stripes of images, functions of items or other domain

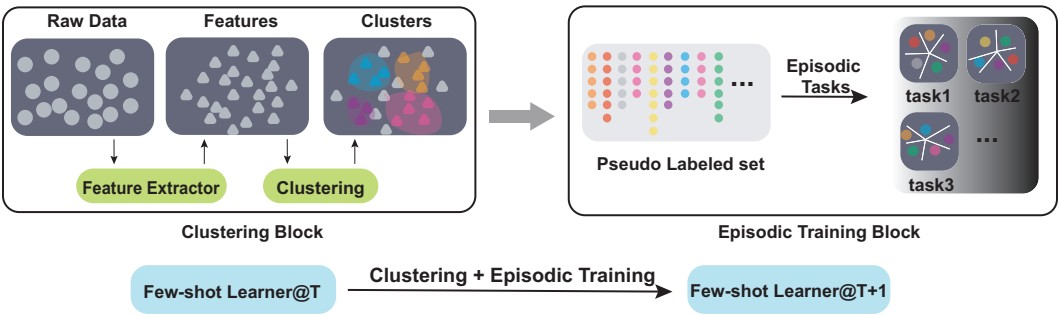

Figure 1: The scheme of our model UFLST, which consists of two alternate processes: clustering and episodic training. At each round, unlabeled data points are clustered based on extracted features, and pseudo labels are assigned according to cluster identities. After clustering, a set of episodic tasks are constructed by sampling from the pseudo-labeled data, and the few-shot learner is trained, which further optimizes feature representations. The two processes are repeated.

knowledge. Furthermore, humans can extract representative representations across categories and apply this capability to learn new concepts (Wang et al., 2014b).

In the present study, inspired by the unsupervised and progressive characteristics of human learning, we propose an unsupervised model for few-shot learning via a self-supervised training procedure (UFLST). Different from previous unsupervised learning methods, our model integrates unsupervised learning and episodic training into a unified framework, which facilitates feature extraction and model training iteratively. Basically, we adopt the episodic training paradigm, taking advantage of its capability of extracting transferable knowledge across tasks, but we use an unsupervised strategy to construct episodic tasks. Specifically, we apply progressive clustering to generate pseudo labels for unlabeled data, and this is done alternatively with feature optimization via few-shot learning in an iterative manner (Fig. 1). Initially, unlabeled data points are assigned into several clusters, and we sample a few training examples from each cluster together with their pseudo labels (the identities of clusters) to construct a set of episodic tasks having the same setting as the testing environment. We then train the few-shot learner using the constructed episodic tasks and obtain improved feature representations for the data. In the next round, we use the improved features to re-cluster data points, generating new pseudo labels and constructing new episodic tasks, and train the few-shot learner again. The above two steps are repeated till a stopping criterion is reached. After training, we expect that the few-shot learner has acquired the transferable knowledge (the optimized feature representations) suitable for a novel task of the same setting as in the episodic training. Using benchmark datasets, we demonstrate that our model outperforms other unsupervised few-shot learning methods and approaches to the performances of fully supervised models.

## 2 RELATED WORKS

In the paradigm of episodic training, few-shot learning algorithms can be divided into two main categories: "learning to optimize" and "learning to compare". The former aims to develop a learning algorithm which can adapt to a new task efficiently using only few labeled examples or with few steps of parameter updating (Finn et al., 2017; Ravi & Larochelle, 2016; Mishra et al., 2017; Rusu et al., 2018; Nichol & Schulman, 2018; Andrychowicz et al., 2016), and the latter aims to learn a proper embedding function, so that prediction is based on the distance (metric) of a novel example to the labeled instances (Vinyals et al., 2016; Snell et al., 2017; Ren et al., 2018; Sung et al., 2018; Liu et al., 2018). In the present study, we focus on the "learning to compare" framework, although the other framework can also be integrated into our model.

Only very recently, people have tried to develop unsupervised few-shot learning models. Hsu et al. (2018) proposed a method called CACTUs, which uses progressive clustering to leverage feature representations before carrying out episodic training. This is different from our model, in term of that we carry out progressive clustering and episodic training concurrently, and the two processes facilitate with each other. Khodadadeh et al. (2018) proposed a method called UMTRA, which uti-

lizes the statistical diversity properties and domain-specific augmentations to generate training and validation data. Antoniou & Storkey (2019) proposed a similar model called AAL, which uses data augmentations of the unlabeled support set to generate the query data. Both methods are different from our model, in term of that we use a progressive clustering strategy to generate pseudo labels for constructing episodic tasks.

The idea of self-supervised training is to artificially generate pseudo labels for unlabeled data, which is useful when supervisory signals are not available or too expensive (de Sa, 1994). Progressive (deep) clustering is a promising method for self-supervised training, which aims to optimize feature representations and pseudo labels (cluster assignments) in an iterative manner. This idea was first applied in NLP tasks, which tries to self-train a two-phase parser-reranker system using unlabeled data (McClosky et al., 2006). Xie et al. (2016) proposed a Deep Embedded Clustering network to jointly learn cluster centers and network parameters. Caron et al. (2018) further proposed strategies to solve the degenerated solution problem in progressive clustering. Fan et al. (2018) and Song et al. (2018) applied the progressive clustering idea to the person re-identification task, both of which aim to transfer the extracted feature representations to an unseen domain. None of these studies have integrated progressive clustering and episodic training in few-shot learning as we do in this work.

## 3 METHOD

In this section, we describe the model UFLST in detail. Let us first introduce some notations. Denote the model at the training round $t$ as $M^t$, $\{x_i\}$ the unlabeled dataset with the number of examples $N$, and $\{z_i\}^t$ is the corresponded feature vector with dimentionality $D$, which is given by $f_{\theta^t} : x \xrightarrow{M^t} z$, where $f_{\theta^t}$ representing the feature extracter and $\theta^t$ the training parameters of $M^t$. $\{\widetilde{z}_i\}^t$ and $\{\widetilde{x}_i\}^t$ represent, respectively, the selected features and unlabeled data after removing outliers from the clustering results, and $\{\widetilde{y}_i\}^t$ the corresponding pseudo labels.

### 3.1 PROGRESSIVE CLUSTERING

#### 3.1.1 K-RECIPROCAL JACCARD DISTANCE

To cluster unlabeled data points, we adopt the k-reciprocal Jaccard distance (KRJD) metric to measure the distance between data points (Qin et al., 2011; Zhong et al., 2017), and it is done in the feature space rather than the raw pixel space. First, we calculate the k-reciprocal nearest neighbours of each feature point, which are given by,

$$R(z, k) = \{z_j \,|\, (z_j \in N(z, k)) \cap (z \in N(z_j, k))\}, \tag{1}$$

where $N(z, k)$ denotes the $k$ nearest neighbours of $z$. $R(z, k)$ imposes that $z$ and each element of $R(z, k)$ are mutually $k$ nearest neighbours of each other. Second, we compute KRJD between two feature points, which is given by

$$J_{ij} = 1 - \frac{|R(z_i, k) \cap R(z_j, k)|}{|R(z_i, k) \cup R(z_j, k)|}. \tag{2}$$

Compared to the Euclidean distance, KRJD takes into account the reciprocal relationship between data points, and hence is a stricter rule measuring whether two feature points matches or not. We find that KRJD is crucial to our model, which outperforms the Euclidean metric as demonstrated in Fig. 2 (more comparisons of two metrics are given in Appendix A).

#### 3.1.2 DENSITY-BASED SPATIAL CLUSTERING ALGORITHM

For the clustering strategy, we choose the density-based spatial clustering algorithm (DBSCAN) (Ester et al., 1996), which performs better than other methods in our model. The reasons are: 1) other clustering methods such as k-means and hierarchical clustering are useful to find spherical shaped or convex clusters in the embedding space, while DBSCAN works well for arbitrarily shaped clusters; 2) DBSCAN can detect outliers as noise points, which is very useful at the beginning of training when the distribution of data points in the embedding space is highly noisy; 3) DBSCAN does not need to specify the number of clusters to be generated, which is appealing for unsupervised learning.

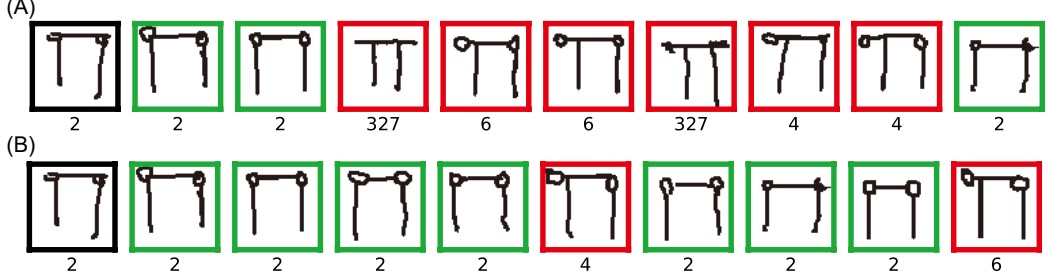

Figure 2: Comparing the performances of KRJD and the Euclidean metric. Top 10 neighbours of a chosen query character from Omniglot are shown. Black box: the query character. Green box: the positive characters in the neighbourhood of the query character. Red box: the negative characters in the neighbourhood of query character. (A) The ranking result using Euclidean metric. (B) The ranking result using KRJD. The number under each image represents its true class. KRJD outperforms the Euclidean metric, in term of it includes more positive examples in the ranking list.

After removing noisy points (outliers) as done in DBSCAN, pseudo labels (i.e., cluster identities) of data points $\{\widetilde{x}_i\}^t$ can be expressed as,

$$\{\widetilde{y}_i\}^t = DBSCAN(\epsilon, ms, J), \tag{3}$$

where $\epsilon$ denotes the maximum distance for two points to be considered as in the same neighborhood, $ms$ the minimum number of points huddled together for a region to be considered as dense, and $J$ the KRJD matrix. The value of $\epsilon$ relies on the cluster density of feature points, which is set to be the mean distance of top $P$ minimum distances in KRJD in this study (see Appendix B).

## 3.2 EPISODIC TRAINING

### 3.2.1 CONSTRUCTING EPISODIC TASKS

After each round of clustering, we construct a number of episodic tasks, denoted as $\mathcal{T} = \{T_1, T_2, ..., T_S\}$ with $S$ the number of tasks, from the pseudo-labeled data set $\{\widetilde{x}_i, \widetilde{y}_i\}^t$. For each episodic task, we randomly sample $N_C$ classes and $N_E$ examples per class. Notably, the setting of each episodic task follows that of the test environment to be performed after training.

We will apply two different ways to implement few-shot learning (see below). One uses the prototype loss, which aims to learn the prototypes of each class and discriminate a novel example based on its distances to the prototypes. In this case, we further split $N_E$ into a support set $S$ and a query set $Q$, i.e., $N_E = N_S + N_Q$. Following Snell et al. (2017), we choose a larger value of $N_C$ in training than that in testing, but keeps $N_S$ the same. The other way to implement few-shot learning is to use the triplet loss or the hardtriplet loss, which separates examples from different classes with a positive margin $m$. In this case, no splitting support and query data is needed. To mine hard negative examples in triplets, we also use a larger $N_C$ in training than that in testing.

### 3.2.2 LOSS FUNCTIONS

Two types of loss functions are used in the present study, and both of them are in the framework of "learning to compare" and contribute to simple inductive bias of our model. One is the prototype loss, which is written as

$$L_{proto}(z, c_p; \theta) = \frac{\exp(-\|z - c_p\|_2^2)}{\sum_k \exp(-\|z - c_k\|_2^2)}, \tag{4}$$

where $c_k$ is the prototype of class $k$ given by $c_k = \sum_{z_i \in S_k}(z_i)/S_k$, and $z$ a query point. In implementation, we choose to minimize the negative log value of Eq. 4, i.e., $L_{proto}^{\log}(z, c_k; \theta) = -\log L_{proto}(z, c_k; \theta)$, as the log value better reflects the geometry of the loss function, making it easy to select a learning rate to minimize the loss function.

---

**Algorithm 1** Unsupervised Few-shot Learning via Self-supervised Training (UFLST)

---

**Input:** Unlabeled dataset $\{x_i\}$, initialized model parameters $\theta^0$
**Output:** Trained model parameters $\theta^T$
 1: $t = 0$
 2: **repeat**
 3:     **Clustering:**
 4:     Extracting features $\{z_i\}^t$ of $\{x_i\}$ using the feature extractor $f_{\theta^t}$.
 5:     Computing K-reciprocal nearest neighbours $R(z_i, k)^t$ of each $z_i$.
 6:     Calculating Jaccard distance matrix $J$ based on $\{R(z_i, k)\}^t$.
 7:     Clustering data using DBSCAN and generating pseudo labels $\{y_i\}^t$.
 8:     Removing outliers and obtaining the pseudo-labeled dataset $\{\widetilde{x}_i, \widetilde{y}_i\}^t$.
 9:     **Episodic Training:**
10:     $s = 0$
11:     **repeat**
12:         Constructing a episodic task $\mathcal{T}^s$ by randomly sampling $N_C$ classes with $N_E$ examples per class from $\{\widetilde{x}_i, \widetilde{y}_i\}^t$.
13:         Updating model parameters $\theta^t$ by training the few-shot learner on $\mathcal{T}^s$.
14:         $s = s + 1$
15:     **until** $s = S$
16:     $t = t + 1$
17: **until** $t = T$

---

The other is the triplet loss (Weinberger & Saul, 2009), which has been widely used in face recognition and image retrieval. The triplet loss $L_{triplet}$ consists of several triplets, each of which includes a query feature $z$, a positive feature $z^+$ and a negative feature $z^-$, and is written as

$$L_{triplet}(z, z^+, z^-; \theta) = \max(0, \|z - z^+\|_2^2 - \|z - z^-\|_2^2 + m), \tag{5}$$

where $m$ controls the margin of two classes, and the hinge term plays the role of correcting triplets, so that the difference between the similarities of positive and negative examples to the query point is larger than a margin $m$. However, in the above form, positive pairs in those "already correct" triplets will no longer be pulled together due to the hard cutoff. We therefore replace the hinge term by a soft-margin formulation, which gives

$$L_{triplet-SM}(z, z^+, z^-; \theta) = \log\left[1 + \exp(\|z - z^+\|_2^2 - \|z - z^-\|_2^2 + m)\right]. \tag{6}$$

Eq. 6 is similar to Eq. 5, but it decays exponentially instead of having a hard cutoff and tends to be numerically more stable (Hermans et al., 2017) (see Appendix C for more details).

We find that in general our model achieves a better performance using the prototype loss than using the triplet loss. However, by including hard example mining when constructing triplets, referred to as the hardtriplet loss hereafter, the model performance is improved significantly and becomes better that using the prototype loss. The relationships between these loss functions are analyzed in Appendix D. The pseudo code of our model is summarized in Algorithm 1.

## 4 EXPERIMENTS

### 4.1 DATASETS

We evaluate our model on two benchmark datasets, which are Omniglot (Lake et al., 2015) and Market1501 (Zheng et al., 2015).

**Omniglot** contains 1623 different handwritten characters from 50 different alphabets. There are 20 examples in each class and each of them was drawn by a different human subject via Amazon's Mechanical Turk. Following Vinyals et al. (2016), we split data into two parts: 1200 characters for training and 423 for testing, but we did not augment data with rotations (this is unnecessary in our model), and instead of resizing images to $28 \times 28$, we resized them to $32 \times 32$.

**Market-1501** is a person re-identification (Re-ID) dataset containing 32668 images with 1501 identities captured from 6 cameras. The dataset is split into three parts: 12936 images with 751 identities

forming the training set, 19732 images with 750 identities forming the gallery set, and another 3368 images forming the query set. All images were resized to $256 \times 128$. Except normalization, no other pre-processing was applied.

## 4.2 IMPLEMENTATION DETAILS

When training on the Omniglot dataset, we chose the model architecture to be the same as that in Vinyals et al. (2016) , which consists of four stacked layers. Each layer comprises 64-filter $3 \times 3$ convolution, followed by a batch normalization, a ReLU nonlinearity, and $2 \times 2$ max-pooling. When training on the market1501 dataset, due to high variances of pose and luminance, we chose to use a highly expressive model (Xiong et al., 2018), which consists of a Resnet50 pretrained on ImageNet as a backbone, and a batch normalization layer after the global max-pooling layer to prevent overfitting. Our evaluation protocol on market1501 is different from that in Zheng et al. (2015), where they reported Cumulative Matching Characteristic (CMC) and the mean average precision (mAPs); while we consider the performance of 1-shot learning, which mimics the typical single query condition in a person Re-ID task.

When training with the triplet loss, we set the margin between positive and negative examples to be 0.5, and the number of training rounds $T = 20$. To avoid overfitting, the model is fine-tuned for only 50 epochs in each round. We used Adam with momentum to update the model parameters, and the learning rate is set to 0.005 with an exponential decay after 25 epochs. The mini-batch size is 128, which consists of 32 classes and 4 examples per class in each episodic task. When constructing triplets with hard example mining, we didn't mine hard negative examples across the whole dataset which is infeasible, rather we did only in the current episodic task. When training with the prototype loss, we used more classes (higher way) during training ($N_C = 60$ in Omniglot and $N_C = 30$ in Market1501), which leads to better performances as empirically observed in Snell et al. (2017). Other hyper-parameters are set to be the same as training with the triplet loss.

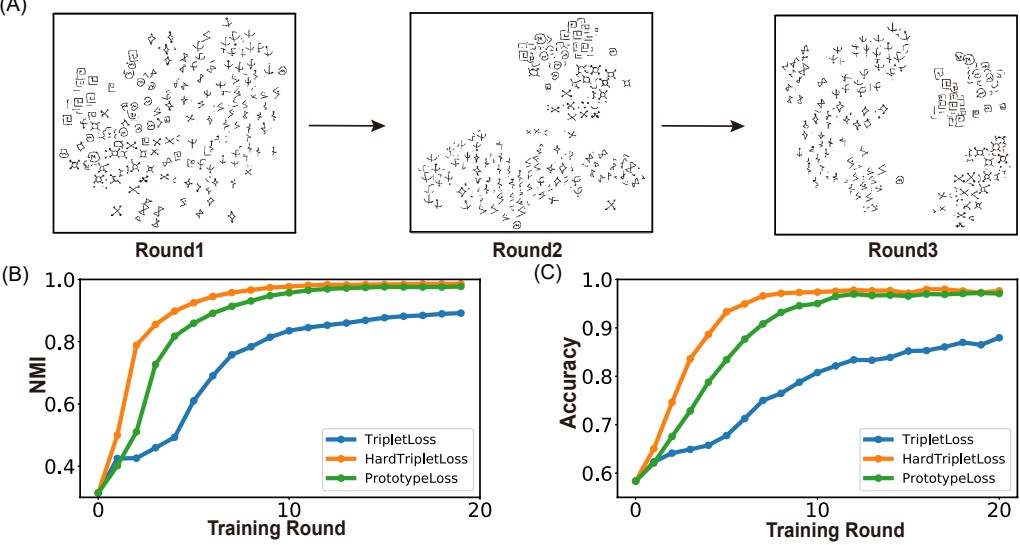

Figure 3: Behaviors of progressive clustering. (A) Visualizing clustering results over training rounds by T-SNE. 10 characters from the Futurama alphabets in the Omniglot dataset were selected. (B) NMI vs. training round. (C) Classification accuracy vs. training round.

## 4.3 PERFORMANCE OF PROGRESSIVE CLUSTERING

We first check the behavior of progressive clustering via visualizing 10 hand-written characters from the Futurama alphabets in the Omniglot dataset using T-SNE (Maaten & Hinton, 2008). Overall, we observe that as learning progresses, the organization of data points is improved continuously, indicating that our model "discovers" the underlying structure of data gradually. As illustrated in Fig. 3A,

initially, all data points are intertwined with each other and no structure exists. Over training, clusters gradually emerge, in the sense of that data points from the same class are grouped together and the margins between different classes are enlarged. Appendix E presents a more detailed illustration of the clustering process. We quantitatively measure the clustering quality by computing the Normalized Mutual Information (NMI) between real labels $\{\hat{y}_i\}^t$ (i.e., the ground truth) and pseudo labels $\{\widetilde{y}_i\}^t$, which is given by,

$$NMI\left(\{\hat{y}_i\}^t, \{\widetilde{y}_i\}^t\right) = \frac{I(\{\hat{y}_i\}^t, \{\widetilde{y}_i\}^t)}{\sqrt{H(\{\hat{y}_i\}^t)H(\{\widetilde{y}_i\}^t)}}, \tag{7}$$

where $I(\cdot, \cdot)$ is the mutual information between $\{\hat{y}_i\}^t$ and $\{\widetilde{y}_i\}^t$, and $H(\cdot)$ the entropy. The value of NMI lies in $[0, 1]$, with 1 standing for perfect alignment between two sets. Note that NMI is independent of the permutation of labeling orders. As shown in Fig. 3B, the value of NMI increases with the training round and gradually reaches to a high value close to 1. Remarkably, the value of NMI well predicts the classification accuracy of the learning model (comparing Fig. 3B and 3C). These results strongly suggest that the combination of progressive clustering and episodic training in our model is able to discover the underlying structure of data manifold and extract the representative features of data points necessary for the few-shot classification task.

## 4.4 RESULTS ON OMNIGLOT

Table 1 presents the performances of our model on the Omniglot dataset compared with other methods. We note that using the triplet loss, our model already outperforms other state-of-the-art unsupervised few-shot learning methods, including CACTUs (Hsu et al., 2018), UMTRA (Khodadadeh et al., 2018), and AAL (Antoniou & Storkey, 2019), to a large extend. Using the prototype loss, the performance of our model is further improved. The best performance of our model is achieved when using the hardtriplet loss. Remarkably, the best performance of our model approaches to that of two supervised models, which are the upper bounds for unsupervised methods.

| | 5-way Acc. | | 20-way Acc. | |
|---|---|---|---|---|
| | 1-shot | 5-shot | 1-shot | 5-shot |
| UMTRA (Khodadadeh et al., 2018) | 77.80 | 92.74 | 62.20 | 77.50 |
| CACTUs-MAML (Hsu et al., 2018) | 68.84 | 87.78 | 48.09 | 73.36 |
| CACTUs-ProtNets (Hsu et al., 2018) | 68.12 | 83.58 | 47.75 | 66.27 |
| AAL-MAML++ (Antoniou & Storkey, 2019) | 88.40 | 97.96 | 70.21 | 88.32 |
| AAL-ProtoNets (Antoniou & Storkey, 2019) | 84.66 | 89.14 | 68.79 | 74.28 |
| UFLST-Tripletloss | 88.68 | 96.65 | 73.21 | 90.11 |
| UFLST-Prototypeloss | 96.51 | **99.23** | 90.27 | 97.22 |
| UFLST-HardTripletloss | **97.03** | 99.19 | **91.28** | **97.37** |
| MAML (Finn et al., 2017) (Supervised) | 98.7 | 99.9 | 95.8 | 98.9 |
| ProtoNets (Snell et al., 2017) (Supervised) | 98.8 | 99.7 | 96.0 | 98.9 |

Table 1: Performances of different unsupervised few-shot learning models on Omniglot under different settings.

## 4.5 RESULTS ON MARKET1501

We also applied our model to a real-world application on person Re-ID. In reality, labeled data is extremely lacking for person Re-ID, and unsupervised learning becomes crucial. Table 2 presents the performances of our model on the benchmark datset Market1501. There is no reported unsupervised few-shot learning result on this dataset in the literature. Rahimpour & Qi (2018) report the supervised results under the 100-way 1-shot scenario. To evaluate our model, we trained a supervised model adapted from Xiong et al. (2018). We find that the model performance using the hardtriplet loss is much better than that using the prototype loss. This is due to that large variations in the appearance and environment of detected pedestrians lead to that noisy samples may be chosen as the prototypes, which deteriorates learning; while the hardtriplet loss focuses on correcting highly noisy examples that violate the margin and hence alleviates the problem. Overall, we observe that our model achieves encouraging performances compared to the supervised method, in particular,

in the scenario of low-way classification, which suggest that our model is feasible in practice for person Re-ID when annotated labels are not unavailable.

|  | 5-way | 10-way | 15-way | 20-way | 50-way | 100-way |
|---|---|---|---|---|---|---|
| UFLST-Tripetloss | 72.8 | 63.0 | 56.2 | 53.4 | 42.5 | 35.4 |
| UFLST-Prototypeloss | 88.3 | 81.2 | 75.8 | 73.0 | 62.5 | 54.0 |
| UFLST-HardTripletloss | **91.4** | **86.9** | **81.6** | **80.4** | **70.1** | **62.1** |
| Our supervised model | 96.8 | 94.7 | 92.5 | 91.1 | 83.7 | 77.3 |
| ARM (Rahimpour & Qi, 2018) | - | - | - | - | - | 76.99 |

Table 2: Performances of our model on Market1501 with different settings. The supervised model is adapted from Xiong et al. (2018). Only 1-shot learning is considered to mimic the typical single query condition in person Re-ID applications.

## 4.6 EFFECT OF THE SIZE OF THE UNLABELED DATASET

We also evaluate how our model relies on the number of unlabeled examples. Table 3 presents the results on the Omniglot dataset with varying number of training examples. Overall, the model performance is improved when the number of training examples increases. Notably, by using only a quarter of the unlabeled data, our model already achieves performances comparable to other unsupervised few-shot learning methods (comparing UFLST-300 with those in Table 1). This demonstrates the feasibility of our model when the number of unlabeled examples is not large.

|  | 5-way Acc. | | 20-way Acc. | |
|---|---|---|---|---|
| Number of Classes | 1-shot | 5-shot | 1-shot | 5-shot |
| UFLST-200 | 82.83 | 92.97 | 65.85 | 83.73 |
| UFLST-300 | 86.03 | 95.05 | 70.52 | 87.60 |
| UFLST-400 | 91.30 | 97.27 | 78.64 | 92.50 |
| UFLST-500 | 95.27 | 98.86 | 87.02 | 96.05 |
| UFLST-1200 | 97.03 | 99.19 | 91.28 | 97.37 |

Table 3: Performances of our model on Omniglot using different numbers of unlabeled training examples. The hardtriplet loss is used.

## 5 DISCUSSION

In this study, we have proposed an unsupervised model UFLST for few-shot learning via self-training. The model consists of two processes, progressive clustering and episodic training, which are executed iteratively. Other unsupervised methods also consider the two processes, but they are performed separately, in term of that unsupervised clustering for feature extraction is accomplished before applying episodic learning. This separation has a shortcoming, since there is no guarantee that the extract features by unsupervised clustering are suitable for the followed few-shot learning. Here, our model carries out the two processes in an alternate manner, which allows them to facilitate with each other, such that feature representation and model generalization are optimized concurrently, and eventually it produces a high quality few-shot learner. To our knowledge, our work is the first one that integrates progressive clustering and episodic training for unsupervised few-shot learning.

On the Omniglot dataset, our model outperforms other state-of-the-art unsupervised few-shot learning methods to large extend and approaches to the performances of supervised modes. On the Market1501 dataset, our model also achieves encouraging performances compared to a supervised method. The high effectiveness of our model makes us think about why it works. Few-shot learning in essence is to extract good representations of data suitable for prediction by using very few training examples. To resolve this challenge, the episodic learning paradigm aims to create a set of episodic few-shot learning scenarios having the same setting as the testing environment, so that the model learns to extract good feature representations that are transferable to novel but related

tasks. To this end, the real labels of data are helpful but not essential, and we can construct pseudo-labeled examples to train the model. But crucially, as demonstrated by this study, the construction of pseudo-labeled examples must go along with the episodic training, so that the extracted features of data really matches the few-shot learning task. Notably, this unsupervised and progressive way of learning agrees with the nature of human on few-shot learning.

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

## A    K-RECIPROCAL JACCARD DISTANCE (KRJD)

A clustering algorithm operating on the raw data space or a shallow linear embedded space is known to be ineffective when the dimensionality of data is high, suffering from the so called "curse of dimensionality" problem (Steinbach et al., 2004). Recent studies on learning representations with deep neural networks have made promising progress for clustering algorithms (Xie et al., 2016; Caron et al., 2018; Yang et al., 2017; Shaham et al., 2018). Although these studies alleviate the problem of choosing data representations, it still needs to choose a suitable distance metric. For some datasets, such as MNIST (LeCun et al., 1998) and STL (Coates et al., 2011), it is adequate to use the Euclidean distance in the feature space to cluster data well. However, for other datasets, such as Omniglot (Lake et al., 2015) and Market1501 (Zheng et al., 2015), where the number of classes is large but the number of examples in each class is small, the distribution of data points in the feature space is quite uneven. In such a case, the Euclidean or Cosine distance does not work well. To solve this problem, we adopt to use the k-reciprocal Jaccard distance (KRJD) metric originally proposed for object retrieval Qin et al. (2011) to measure the distance between two data points..

The Jaccard distance compares the members of two sets to judge which elements are shared or distinct. For two sets $A$ and $B$, the Jaccard distance is calculated to be,

$$J(A, B) = \frac{|A \cap B|}{|A \cup B|}, \tag{8}$$

where the numerator is the number of elements shared by two sets, and the denominator the number of elements in either set. The Jaccard distance is a measurement of the similarity between two sets of data and is in the range of $[0, 1]$, where the lower its value, the more similar two sets are.

The Jaccard distance is originally used for two sets with discrete elements. To apply to a group of data points in the feature space, we adopt KRJD from Zhong et al. (2017). Denote the group of data points as $Z$, where each row is the feature of a data point. KRJD transfers $Z$ into a discrete k-reciprocal ranking list $R$ by first calculating the $k$ nearest neighbours (KNN) of each element in $Z$ and then re-ranking them. Once $R$ is available, we apply Eq. 8 to calculate the Jaccard distance of $Z$. Denote the $k$ nearest neighbours of the feature point $z_i$ to be

$$N(z_i, k) = \{z_i^0, z_i^1, ..., z_i^k\}. \tag{9}$$

Then, the k-reciprocal nearest neighbours (KRNN) of $z_i$ is obtained by re-ranking the KNN list, which is written as,

$$R(z_i, k) = \{z_j \,|(z_j \in N(z_i, k)) \cap (z_i \in N(z_j, k))\}. \tag{10}$$

Compared to KNN, KRNN is a more strict and accurate measurement of ranking, so that more positive examples appear in the top $k$ ranking list, as demonstrated in Fig. 4 and Fig. 5. After obtaining KRNN of each point, we calculate KRJD between two points by

$$J_{ij} = 1 - \frac{|R(z_i, k) \cap R(z_j, k)|}{|R(z_i, k) \cup R(z_j, k)|}. \tag{11}$$

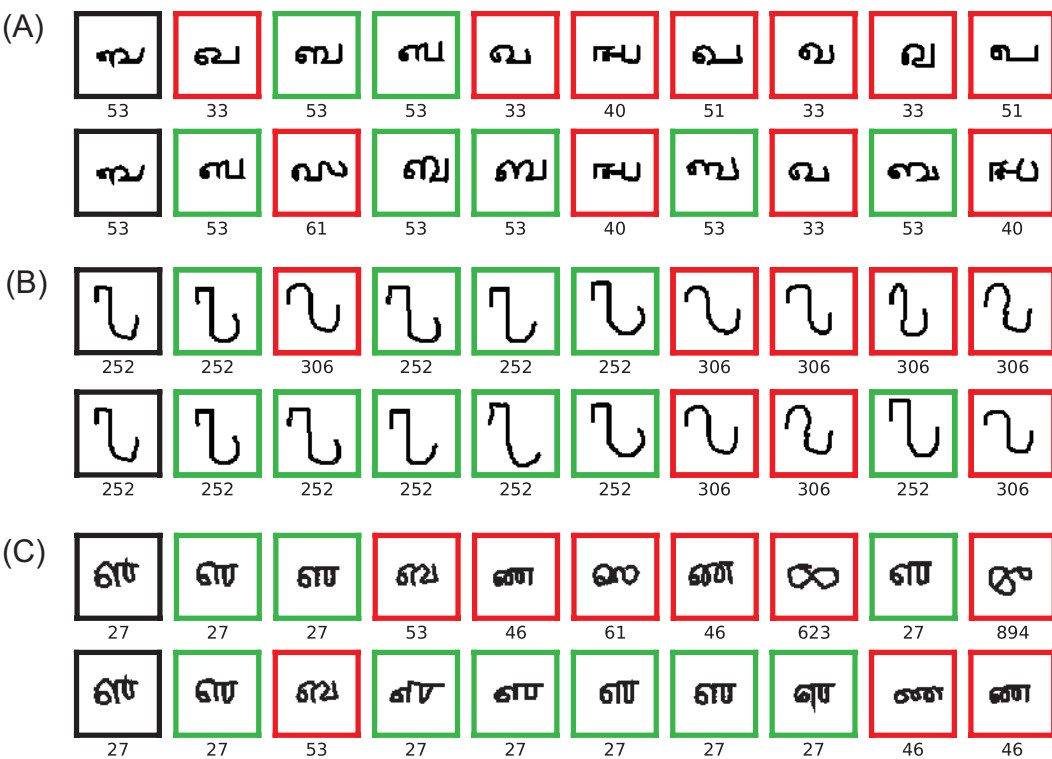

Figure 4: Top 10 neighbours of a chosen query character from Omniglot dataset. Black box: the query character. Green box: the positive characters in the neighbourhood of the query character. Red box: the negative characters in the neighbourhood of query character. The number under each image represents its true class. Upper panels in (A), (B) and (C): the ranking result using the Euclidean metric. Lower panels in (A), (B) and (C): the ranking result using KRJD. The KRJD metric outperforms the Euclidean metric, in term of it includes more positive examples in the ranking list.

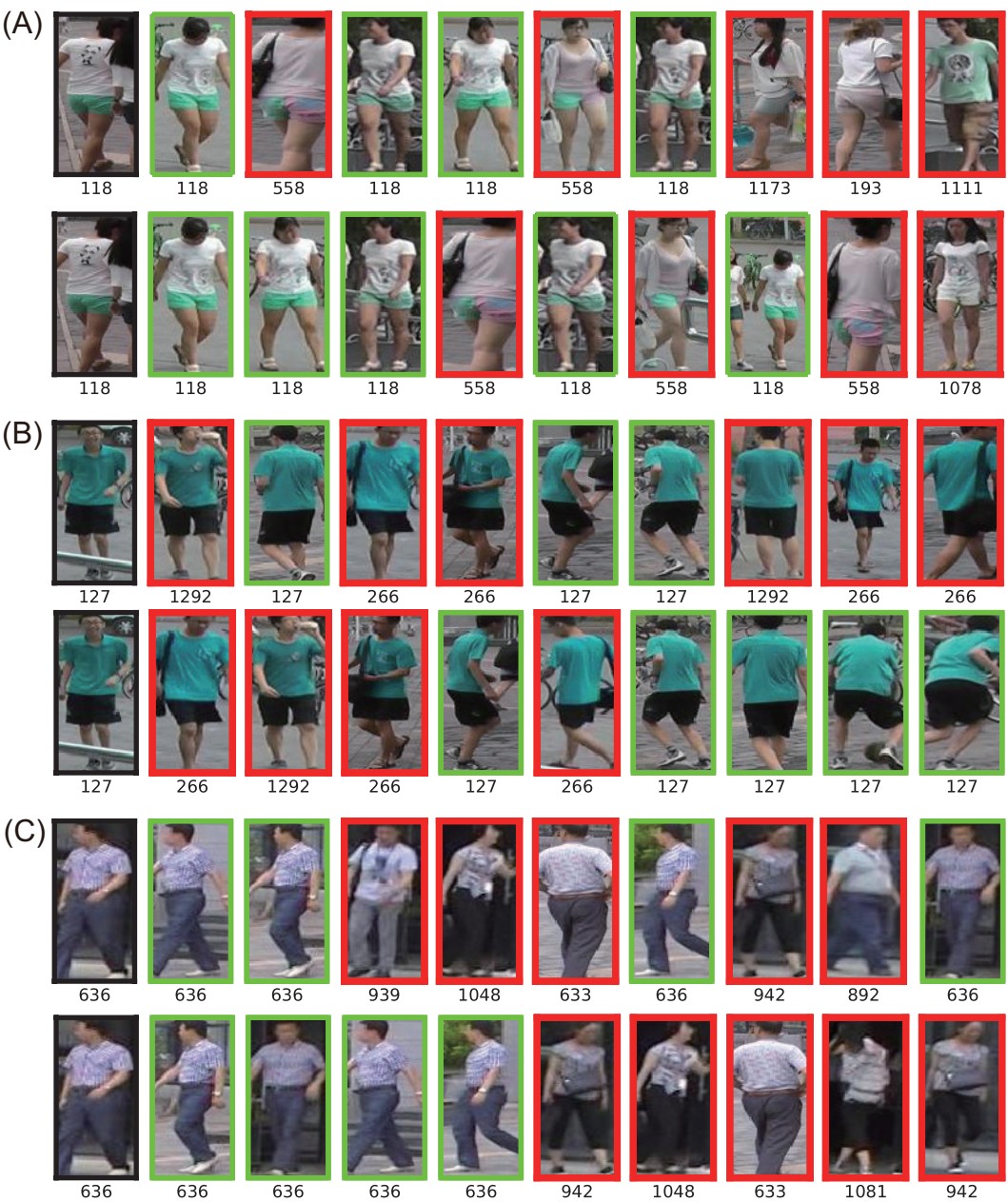

Figure 5: Top 10 neighbours of a chosen query image from the Market1501 dataset. Black box: the query image. Green box: the positive images in the neighbourhood of the query image. Red box: the negative images in the neighbourhood of query image. The number under each image represents its true class. Upper panels in (A), (B) and (C): the ranking results using the Euclidean metric. Lower panels in (A), (B) and (C): the ranking results using KRJD.

## B    THE PARAMETER OF $\epsilon$ IN DBSCAN

The parameter $\epsilon$ in DBSCAN specifies how close two points should be for them to be considered as in the same cluster, and its value relies heavily on the underlying cluster density of feature points. In the present study, we choose $\epsilon$ to be the mean of top $P$ minimum distances of the KRJD matrix $J$, which is given by,

$$\epsilon = \frac{\sum top(J, P)}{P}. \tag{12}$$

To determine the value of $P$, we set $P = \rho |\{\widetilde{x}_i\}|$, where $\rho$ is a hyper-parameter representing the portion of points selected over the whole set $\{\widetilde{x}_i\}$. Increasing $\rho$ equals to increasing $\epsilon$, which implies that more points will be assigned into a single cluster, and consequently this decreases the number of clusters obtained. Decreasing $\rho$ has the opposite effect.

Table 4 presents the effects of different values of $\rho$ on the model performances. We see that when $\rho$ is small, the performance is not good, since there are too many clusters and the number of examples in each cluster is small, which makes it hard to construct episodic tasks having enough examples to train the model; when the value of $\rho$ is too large, the model performance is also bad, since in this case, it is hard to generate sufficiently different episodic tasks. For large $\rho$, we may reduce the value of $N_C$ to alleviate the problem (note that in our model, the learning scenario affects the clustering behavior).

| | 5-way Acc. | | 20-way Acc. | |
|---|---|---|---|---|
| | 1-shot | 5-shot | 1-shot | 5-shot |
| $\rho = 1 \times 10^{-5}, N_C = 60$ | 78.88 | 92.73 | 59.49 | 81.50 |
| $\rho = 2 \times 10^{-5}, N_C = 60$ | 82.85 | 94.82 | 64.83 | 85.61 |
| $\rho = 3 \times 10^{-5}, N_C = 60$ | 91.43 | 97.79 | 78.48 | 93.04 |
| $\rho = 4 \times 10^{-5}, N_C = 60$ | 97.57 | **99.35** | 92.62 | 97.82 |
| $\rho = 5 \times 10^{-5}, N_C = 60$ | 97.03 | 99.19 | 91.28 | 97.37 |
| $\rho = 6 \times 10^{-5}, N_C - 60$ | **97.64** | 99.34 | **93.39** | **98.06** |
| $\rho = 7 \times 10^{-5}, N_C = 60$ | 58.35 | 79.24 | 34.17 | 58.33 |
| $\rho = 8 \times 10^{-5}, N_C = 60$ | 58.10 | 78.95 | 33.90 | 58.12 |
| $\rho = 7 \times 10^{-5}, N_C = 30$ | 97.16 | 99.12 | 91.28 | 97.48 |
| $\rho = 8 \times 10^{-5}, N_C = 20$ | 97.49 | 99.25 | 92.14 | 97.72 |

Table 4: Effects of the $\rho$ value in DBSCAN on model performances. The hyper-parameters are given in Sec. 4.2.

## C    COMPARING THE TRIPLET LOSS AND THE SOFT-MARGIN TRIPLET LOSS

In Sec. 3.2.2, we propose to use the soft-margin version of the triplet loss to replace the conventional one with the hard cutoff. This concerns that when the hard cutoff triplet loss stops to pull together those "already correct" positive pairs, the soft-margin triplet loss continues to optimize those pairs. We also empirically find that the soft-margin triplet loss tends to be numerically more stable during training. Table 5 presents the model performances using the two loss functions.

## D    RELATIONSHIPS BETWEEN THE LOSS FUNCTIONS

In Sec. 4.4 and 4.5, we observe that the model trained with the prototype loss has a better performance than that with triplet loss, and by mining hard examples in the triplets, using the hardtriplet loss, the model achieves the best performance. Here, we give analysis on the relationships between these loss functions.

| | 5-way Acc. | | 20-way Acc. | |
| --- | --- | --- | --- | --- |
| | 1-shot | 5-shot | 1-shot | 5-shot |
| **Omniglot** | | | | |
| Hard-cutoff(m=0.5) | 95.22 | 99.00 | 88.14 | 96.48 |
| Hard-cutpff(m=1.0) | 96.44 | 99.15 | 89.80 | 96.99 |
| Soft-margin(m=0.5) | 96.60 | 99.15 | 90.19 | 97.14 |
| Soft-margin(m=1.0) | **97.03** | **99.19** | **91.28** | **97.37** |
| **Market1501** | | | | |
| Hard-cutoff(m=0.5) | 91.00 | 98.07 | 77.84 | 93.88 |
| Hard-cutoff(m=1.0) | 91.91 | 98.10 | 79.70 | 93.78 |
| Soft-margin(m=0.5) | **93.70** | **98.49** | **82.70** | **95.00** |
| Soft-margin(m=1.0) | 92.34 | 98.32 | 80.95 | 94.41 |

Table 5: Comparison of the soft-margin triplet loss and the hard cutoff triplet loss on two datasets with margin $m = 0.5$ and $m = 1.0$.

Consider a $N_C$-way 1-shot episodic learning scenario, where a prototype $c_k$ is the support point $z_k$, the prototypical loss is written as

$$
\begin{aligned}
L_{proto}^{\log}(z, z_k; \theta) &= -\log \frac{\exp(-\|z - z_p\|_2^2)}{\sum_k \exp(-\|z - z_k\|_2^2)}, \\
&= -\log \frac{1}{1 + \sum_{k \neq p} \exp(\|z - z_p\|_2^2 - \|z - z_k\|_2^2)}, \\
&= \log \left[ 1 + \sum_{k \neq p} \exp(\|z - z_p\|_2^2 - \|z - z_k\|_2^2) \right].
\end{aligned}
\tag{13}
$$

During training, we construct each episodic task by randomly sampling $N_C$ classes and $N_E$ examples per class. When using the prototype loss (Eq. 13), a query point $z$ is pulled towards the corresponded support point $z_p$, and meanwhile, $z$ is pushed away from all other support points $\{z_k\}_{k \neq p}$; whereas, when using the triplet soft-margin loss (Eq. 6), the query point $z$ is only pushed away from one of other support points $z^-$. This implies that in each update, $L_{triplet-SM}$ only interacts with a single negative example from one of other classes and ignores many other negative examples. When $N_C$ is small, optimizing the model with the two loss functions has no big difference. For example, when $N_C = 2$ and $m = 0$, Eq. 6 and 13 become exactly the same. However, when $N_C$ becomes larger, the possible number of triplets grows cubically with $N_E$ and linearly with $N_C$, which makes it difficult to select non-trivial triplets. In such a situation, optimizing on these uninformative triplets leads to the problem that the model gets stuck into a local optimum and suffers slow convergence. This justifies why the model has a inferior performance using the triplet loss compared to using the prototype loss.

The inefficiency of the conventional triplet loss motivate us to mine hard triplets to alleviate its shortcomings (Wang et al., 2014a; Cui et al., 2016; Hermans et al., 2017). Mining hard negative examples across the whole dataset is infeasible, since it is too time-costing to evaluate all embedding vectors in the deep learning framework. So, we choose to do hard negative example mining within a batch, i.e., we select the hardest positive and the hardest negative examples when forming the triplets, and obtain

$$
L_{triplet-SM}^{hard} = \log \left[ 1 + \exp(\max_{z_p \in \{z^+\}} \|z - z_p\|_2^2 - \min_{z_n \in \{z^-\}} \|z - z_n\|_2^2 + m) \right].
\tag{14}
$$

Compared to Eq. 13 which pushes a query point away from all other support points from different classes, Eq. 14 focuses on pulling the hardest positive example closer and pushing the hardest negative example away at the same time. By this, we get a better performance than that using the prototype loss.

Another improvement can be made by using $m > 0$, as a positive margin makes different classes become more separable. At the beginning of training, when data points are intertwined with each other, a positive margin push points belonging to different pseudo classes away quicker than $m = 0$. Table 6 shows the effect of $m$ value on the model performance.

| margin | 5-way Acc. | | 20-way Acc. | |
|---|---|---|---|---|
| | 1-shot | 5-shot | 1-shot | 5-shot |
| $m = 0$ | 96.59 | 99.14 | 90.35 | 97.11 |
| $m = 0.3$ | 96.09 | 98.95 | 89.43 | 96.65 |
| $m = 0.5$ | 96.60 | 99.15 | 90.19 | 97.14 |
| $m = 1.0$ | **97.03** | **99.19** | **91.28** | **97.37** |
| $m = 3.0$ | 96.00 | 98.96 | 88.30 | 96.56 |

Table 6: The model performances using the hardtriplet loss with different $m$ values on Omniglot.

## E   CLUSTERING BEHAVIORS OF DBSCAN

Following Fig. 3 in the main text, we further illustrate the clustering behavior of DBSCAN on Omniglot in more detail. As shown in Fig. 6, at the first few rounds of learning, feature points $\{z_i\}^t$ are unstructured and noisy, they are intertwined with each other, and DBSCAN tends to cluster all data points into a few dominant clusters. Along with learning, the number of clusters keeps increasing, and the number of data points in each cluster becomes more evenly distributed. Eventually, the number of clusters generated at the final training round is around 1100, which is close to the the real number of 1200 in the Ominglot dataset, indicating that our model, which combining progressive clustering and episodic learning, is able to discover the underlying structure of the data manifold.

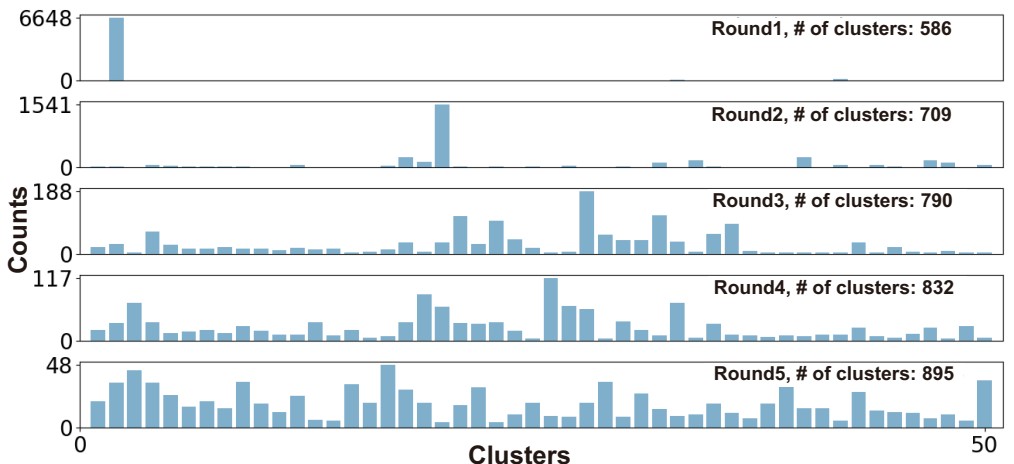

Figure 6: Clusters generated by DBSCAN over training rounds. The results of the first 5 rounds are shown. For the convenience of visualization, we only display 50 clusters and the number of data points in each of them.

## F   THE FREQUENCY OF CLUSTERING DURING TRAINING

| Number of Update Epochs | 1 | 2 | 5 | 10 | 20 | 30 | 40 |
|---|---|---|---|---|---|---|---|
| UFLST-Tripetloss | 97.43 | 97.51 | 96.91 | 97.03 | 97.44 | 97.80 | 97.93 |

Table 7: The frequency of clustering during training. Experiments are carried on the Omniglot dataset. "Number of Update Epochs" means how many epochs are taken before the clustering. Experiments are carried out on Omniglot dataset, under the 5-way 1-shot learning scenario.

Here we investigate the effect of clustering frequency during training. That is, how many epochs are taken before updating the clustering in each round? We find that our model is very robust to

the frequency term. We get 97.43% accuracy under the 5-way 1-shot scenario when the model is trained with an extremely low clustering frequency value, e.g., only trained with one epoch before clustering. Also, we get 97.93% when trained with 40 epoch before clustering (Table 7).

## G    RESULTS ON MINI-IMAGENET

|  | (5,1) | (5,5) | (5,20) | (5, 50) |
|---|---|---|---|---|
| Training fram scratch | 25.17 | 33.90 | 39.56 | 41.45 |
| BiGAN knn-nearest neighbors | 25.56 | 31.10 | 37.31 | 43.60 |
| BiGAN linear classifier | 27.08 | 33.91 | 44.00 | 50.41 |
| BiGAN MLP with dropout | 22.91 | 29.06 | 40.06 | 48.36 |
| BiGAN cluster matching | 24.63 | 29.49 | 33.89 | 36.13 |
| BiGAN CACTUs MAML | 36.24 | 51.28 | 61.33 | 66.91 |
| BiGAN CACTUs ProtoNets | 36.62 | 50.16 | 59.56 | 63.27 |
| DeepCluster knn-nearest neighbors | 28.90 | 42.25 | 56.44 | 63.90 |
| DeepCluster linear classifier | 29.44 | 39.79 | 56.19 | 65.28 |
| DeepCluster MLP with dropout | 29.03 | 39.67 | 52.71 | 60.95 |
| DeepCluster cluster matching | 22.20 | 23.50 | 24.97 | 26.87 |
| DeepCluster CACTUs MAML | 39.90 | **53.97** | **63.84** | **69.64** |
| DeepCluster CACTUs ProtoNets | **39.18** | 53.36 | 61.54 | 63.55 |
| UMTRA without data Augmentation | 26.49 | - | - | - |
| UMTRA+Shift+random flip | 30.16 | - | - | - |
| UMTRA+Shift+random flip +randomly change to grayscale | 32.80 | - | - | - |
| UMTRA+Shift+random flip+random rotation+color distortions | 35.09 | - | - | - |
| UMTRA+AutoAugment | **39.93** | **50.73** | 61.11 | 67.15 |
| AAL-MAML+++ CHV | 33.06 | 40.75 | - | - |
| AAL-MAML+++ CHVR | 33.21 | 40.34 | - | - |
| AAL-MAML+++ CHV + CUT | 33.34 | 39.44 | - | - |
| AAL-MAML+++ CHV + DROP | 30.86 | 40.41 | - | - |
| AAL-MAML+++ CHVW | 33.30 | 46.98 | - | - |
| AAL-MAML+++ CHVWG | **34.57** | **49.18** | - | - |
| AAL-MAML+++ CHVR + CUT | 33.09 | 40.11 | - | - |
| AAL-MAML+++ CHVR + DROP | 31.70 | 39.38 | - | - |
| AAL-MAML+++ CHV + DROP + CUT | 31.55 | 38.76 | - | - |
| AAL-MAML+++ CHVR + DROP + CUT | 31.44 | 39.87 | - | - |
| AAL-ProtoNets+ CHV | **37.67** | **40.29** | - | - |
| AAL-ProtoNets+ CHV + CUT | 36.38 | 40.89 | - | - |
| AAL-ProtoNets+ CHV + CUT + DROP | 33.13 | 36.64 | - | - |
| AAL-ProtoNets+ CHVR + CUT + DROP | 31.93 | 36.45 | - | - |
| AAL-ProtoNets+ CHVR + CUT | 33.92 | 39.87 | - | - |
| AAL-ProtoNets+ CHV + DROP | 32.12 | 36.12 | - | - |
| AAL-ProtoNets+ CHVR + DROP | 31.13 | 36.83 | - | - |
| AAL-ProtoNets+ CHVR | 34.28 | 39.83 | - | - |
| UFLST without data Augmentation | 33.77 | 45.03 | 53.35 | 56.72 |
| MAML (Finn et al., 2017) (Supervised) | 46.81 | 62.13 | 71.03 | 75.54 |
| ProtoNets (Snell et al., 2017) (Supervised) | 46.56 | 62.29 | 70.05 | 72.04 |

Table 8: Performances of different unsupervised few-shot learning models on Mini-ImageNet under different settings. The accuracy with std of our model is :33.77% ± 0.70%, 45.03% ± 0.73%, 53.35% ± 0.59%, 56.72% ± 0.67% on 5-way 1-shot, 5-way 5-shot, 5-way 20-shot, 5-way 50-shot, respectively.

Overall, training a few shot learner on the Mini-ImageNet dataset under the unsupervised setting is very tricky. All the three aforementioned approaches adopt domain specific knowledge and data augmentation tricks in their training. For example, UMTRA uses the statistical likelihood of picking different classes for the training data of $T_i$ in case of $K = 1$ and large number of classes, and an augmentation function $T$ fors the validation data. CACTUs relies on an unsupervised feature

learning algorithm to provide a statistical likelihood of difference and sameness in the training and validation data of $T_i$. The choice of the right augmentation function for UMTRA and AAL, the right feature embedding approach for CACTUs, and the other hyper-parameters have a strong impact on the performance.

The model architecture trained on the Mini-ImageNet dataset is exactly the same as on the Omniglot dataset, i.e., the 4-layer convnet described in Sec.4.2. We only report the results by training without any data augmentation. We achieve $33.77\%$ and $44.03\%$ under the 5-way 1-shot and 5-way 5-shot scenario respectively. Compare to the model training from scratch ($25.17\%$ under the 5-way 1-shot scenario), our model has a gain of $8.6\%$. The best 5-way 1-shot accuracy in the CACTUs model is $39.18\%$. However, comparing to the CACTUs model is unfair because they used the AlexNet or the VGG16 to first learn a very good feature embedder for downstream feature clustering process, while our model is only composed of a 4-layer convenet. Both of the best results in the UMTRA model and the ALL model are acquired by using fancy data augmentations, such as shifting, random flipping, color distortions, image-Warping and image-pixel dropout (see Khodadadeh et al. (2018); Antoniou & Storkey (2019) for more details) while we don't use any data augmentation tricks here. It is noteworthy that our model outperforms the UMTRA trained without any data augmentation to a large extent ($33.77\%$ vs. $26.49\%$).

Compared to the results on Omniglot and Market1501, the results on the Mini-ImageNet is not the state-of-the-art. The underline reason may come from three aspects. (1) For a fair comparison to other unsupervised few-shot learning models, we use the 4-layer convnet. However, the in-class variations of the Mini-ImageNet is very large, which is hard for such a small network to capture the semantic meanings of images. (2) In unsupervised learning, it is hard to choose suitable hyper-parameters, such as the clustering frequency, DBSCAN-related parameters, and the learning rate. (3) The ground truth for the class number of Mini-ImageNet is small 1(64 for training, 16 for validating and 20 for testing). But, for constructing episodic tasks, we prefer to over-segment the dataset, and this over-segmentation tend to assign data belonging to the same class into different clusters, leading to a degenerate performance. Our model performs very well on Omniglot and Maket1501, which may be attributed to that both datasets have large class numbers and the number of examples in each class is small. This type of dataset is very suitable for constructing episodic tasks to learn a few-shot learner. In our future work, we will explore more domain specific knowledge and data augmentation strategies to improve the accuracy on the MiniImageNet dataset and extend our model to more datasets.

