# OpenReview forum: "Unsupervised Few Shot Learning via Self-supervised Training"
_ICLR.cc/2020/Conference — Reject_

### Official Review · AnonReviewer2 · 2019-10-21
**Official Blind Review #2**

**Rating:** 3

**Review:**

This paper considers the problem of learning an image representation for few-shot learning without using image labels during training. This is a well-motivated problem since (as the paper points out) learning such a representation using "episodes" of low-shot learning problems as examples may require a large amount of annotated data. The paper proposes an iterative algorithm which alternates between clustering the images using the current model and updating the model using the clusters as "pseudo-labels". This approach is not particularly elegant as there is no clear objective being optimized, but it may nevertheless be effective. One main claim of the paper is that the iterative nature of this process is key to the success of the algorithm. The choice of model is a multi-layer conv-net which is trained using SGD (Adam). The paper investigates both triplet (with and without hard negatives) and "prototype" losses for learning the model parameters. To find clusters, the paper adopts the DBSCAN algorithm using the Jaccard similarity of the k-reciprocal neighbour sets.

I am not aware of other papers that use self-supervised learning to obtain a representation which is specifically suitable for few-shot learning via nearest-neighbour classification. As is noted in the paper, the absence of supervision during training more closely resembles the scenario of few-shot learning in biological systems.

The design decisions are well motivated throughout the paper.

The appendices are high quality and make the paper much more complete. In particular: the method for choosing epsilon, the empirical study of the effect of epsilon and the discussion of the behaviour of the cluster sizes as training proceeds.

Principal concerns:

(1.1) The proposed approach is particularly similar to DeepCluster (Caron et al.). Besides the use of a different clustering algorithm, it seems that the main high-level difference is the use of "episodic training", in which the algorithm is trained to compare examples to a query example, rather than to classify single examples. I would have preferred to see a comparison to non-episodic training. (It might be necessary to train a linear classifier on top of the final feature representation rather than simply build a nearest-neighbour classifier, but still this is convex, cheap and even closed-form in the case of least-squares regression.)

(1.2) While the algorithm has been demonstrated on real images in the Market1501 dataset, it would have been much more convincing to see it demonstrated on a more widely-used dataset for few-shot learning such as Mini-ImageNet.

(1.3) There are several recent papers on unsupervised feature learning using self supervision, especially as an alternative to ImageNet-classification pre-training. There is no discussion of these approaches, yet they might perform better than the proposed algorithm, especially for tasks with real images. Some examples of such papers are:
- "Discriminative Unsupervised Feature Learning with Exemplar Convolutional Neural Networks"
- "Representation Learning with Contrastive Predictive Coding"
- "Unsupervised representation learning by predicting image rotations"
The literature on using auto-encoders to learn feature representations is also relevant.

Issues with details in the paper:

(2.1) The DeepCluster paper observed that non-negligible accuracy could be achieved using a randomly-initialized conv-net (12% when chance is 0.1%). This enabled the use of clusters as pseudo-labels. Is a similar effect observed with your datasets and random initialization?

(2.2) For large datasets, the clustering algorithm might be prohibitively expensive? It would be useful to discuss the complexity of this algorithm.

(2.3) It would be interesting to see the effect of varying the frequency of the clustering during training (i.e. how many gradient steps are taken before updating the clustering).

(2.4) It is concerning that the accuracy drops sharply as \rho increases [6, 7, 8]*10^-5 in Table 4.

(2.5) It is stated that batch-normalization at the network output helps prevent over-fitting. Why? My intuition is that it would be more helpful for avoiding regions of the loss function which have a small gradient magnitude. What happens if you remove it?

(2.6) What is the test time procedure? Do you use the mean feature when there are k > 1 shots (even for the network trained with the hinge loss)? Do you L2-normalize the representation vectors?

(2.7) It seems potentially brittle to use hard negatives in the triplet loss with pseudo-labels? If the labels were wrong, then the hard negatives might not really be negatives. Nevertheless, this does not seem to be an issue, at least with these datasets.

(2.8) There are no error-bars anywhere in the paper.

Minor comments:

(3.1) Tables 3 and 4 would be easier to interpret if plotted on an axis.

(3.2) The word "concurrently" suggests that the clustering and the training are performed simultaneously. I would prefer "alternating".

(3.3) A grammatical review of the paper is required. For example in the abstract: "to a large extent (extend) and approaches the performance of (to the performances of) supervised methods".

(3.4) Remember to use \log and \max in latex to improve the appearance.

**Experience Assessment:**

I have published one or two papers in this area.

**Review Assessment: Checking Correctness Of Derivations And Theory:**

I assessed the sensibility of the derivations and theory.

**Review Assessment: Checking Correctness Of Experiments:**

I carefully checked the experiments.

**Review Assessment: Thoroughness In Paper Reading:**

I read the paper at least twice and used my best judgement in assessing the paper.

---

> ### Author Response · Authors · 2019-11-15
> **Our model is trained to do unsupervised few shot learning while the works you mentioned are for unsupervised feature learning.**
>
> Thank you very much for your insightful and constructive comments.
>
> For your principle concerns:
>
> 1.1) The idea of progressive clustering can be traced back to the 1990s [1] and it has widely been used to learn semantic features when no label information is available (reviewed in the related work). The main contribution of Caron et al. (DeepCluster) is proposing a strategy to avoid trivial solutions by re-assigning data points to empty clusters and re-weighting the loss function by the inverse of cluster size. Here, we adopt a different density-based clustering method to assign clusters without needing to pre-specify the number of clusters, i.e., there is no empty cluster (see Sec.3.1.2 in our paper). A more interesting method can be found in [2], where the cluster centroids and the features extractor are optimized jointly. In fact, any clustering-based method can be integrated into our model as long as they can generate pseudo labels for constructing episodic tasks. We would like to point out that the key to our model is the iterative integration of unsupervised clustering and episodic learning. Our results demonstrate that through performing two operations iteratively, the model can acquire the capability of transferring knowledge to facilitate learning from a few exemplars, and this is done in an unsupervised manner.
> The results for non-episodic training were reviewed in [3] (see Table 1 in the paper), where different non-episodic training methods were discussed, including the k-nearest neighbors, the linear classifier, the MLP with dropout, and the clustering matching. While their accuracies for 5-way 1-shot learning on Omniglot are 57.46%, 61.08%, 51.95%, and 54.94% respectively, we get a much better result of 97.03%.
>
> 1.2) We have added the results on the Mini-ImageNet dataset, please see the "General Response". For the detailed comparison to other unsupervised few-shot learning methods, please see Appendix G.
>
> 1.3) Indeed, many methods can be used for unsupervised feature learning (e.g., the prediction-based method, GAN-based method and so on). However, using these methods for unsupervised few-shot learning has two issues: (1) their learning objectives are different from few-shot learning's objective; (2) unsupervised feature learning and episodic learning for few-shot learning are implemented separately, (which is exactly what Hsu et al. have done (the CACTUs model) in their ICLR paper [3]). Here, the key difference of our model is that we integrate them together and perform them iteratively. We demonstrate that this is a much better way of learning a few-shot learner in the unsupervised setting. More specifically, in our model, the feature learning is done by episodic training on the generated pseudo labels, and the learned features (which are improved compared to the previous ones) are then used to generate the next round clusters. The two steps are carried out in an iterative way. This helps our model achieve much better performance than others under the 5-way 1-shot learning scenario on Omniglot (see Table 1).
>
> For your issues with details in our papers:
>
> 2.1) Yes, a randomly initialized 4-layer convnet has the initial accuracy of 52.48% under the 5-way 1-shot learning scenario on Omniglot. We used a Resnet50 pre-trained on ImageNet as the initialized model to train the Market1501, which has the initial accuracy of 48.84% under the 5-way 1-shot learning scenario. We also train our model without pre-training on ImageNet and find that it requires more training rounds to achieve the same accuracy as it does with pre-training. We think that this non-chance level initial accuracy is important for the model.
>
> 2.2) Yes, for a large dataset, the clustering algorithm is prohibitively expensive. Computing the KRJD matrix is the most time-consuming part, whose computational complexity is O(N^3). A solution is to perform KRJD in the k-nearest neighbor of each point, rather than between every two points.
>
> 2.3) See Appendix F in the revised version. We find that our model is very robust to the effects of clustering frequency.
>
> 2.4) In Appendix B, we have explained this and we reduced the value of $N_c$ to solve this issue.
>
> 2.5) The BN layer balances each dimension of the features to make them Gaussian distributed near the surface of the hypersphere. In this way, features from the testing and training data are more likely to appear at the same location in the feature space. Removing it will lead to a slight accuracy decrease (91.4% to 89.9% under 5-way 1-shot on Market1501).
>
> 2.6) Yes, we use the mean feature when there are k>1 shots during testing. We did not L2-normalize the representation features.

---

### Official Review · AnonReviewer1 · 2019-10-23
**Official Blind Review #1**

**Rating:** 1

**Review:**

This paper aims to conduct few-shot learning on unlabeled data (instead of on training tasks with few-shot labeled data per task). The proposed algorithm is a trivial combination of existing clustering method and a few-shot learning method, i.e., the clustering provides pseudo labels, from which a series of few-shot training tasks are generated, and then traditional few-shot learning method can be applied afterward. Many existing techniques are integrated, e.g., k-reciprocal Jaccard distance, DBSCAN clustering, prototypical network, triplet loss, etc. The paper reported the experimental results on Omniglot and Market1501.

The paper suffers from several drawbacks: 1) lack of novelty and originality; 2) problem setting is not convincing enough; 3) only a comparison on a very easy and small dataset (Omniglot) is shown, while the comparison on Market1501 is missing; 4) lack of comparison and discussion of other related works.

Detailed comments:

1. A simple combination of two existing techniques (clustering followed by few-shot learning) without any in-depth analysis cannot be justified as a reasonable contribution for an ICLR paper.

2. The unsupervised few-shot learning setting does not make much sense in practice: we can usually collect many labeled data to generate few-shot learning tasks since these labeled data is not required to be drawn from the same distribution as the target few-shot task (e.g., the test tasks). For example, for few-shot image classification, we can always generate training tasks from ImageNet data. In practice, we only suffer from few-shot labeled data on test tasks. Assuming that none of labeled data is available can lead to unnecessary degrade on few-shot learning performance. The authors need to present more evidence or practical application scenarios to support the practical value of this problem setting.

3. Omniglot is too easy and too small for most of the recent few-shot learning methods. More challenging datasets such as ImageNet or at least its easier subsets (e.g., miniImageNet, tiredImageNet) should be considered.

4. No comparison to any baseline is shown on the other dataset Market1501.

5. More related works need to be compared and discussed:

-Constructing learning tasks via clustering has been studies in “Hsu, K., Levine, S. and Finn, C., 2018. Unsupervised learning via meta-learning. arXiv preprint arXiv:1810.02334.” This paper shares the same idea when constructing the unsupervised episode training tasks.

-Self-supervised techniques to boost the performance of few-shot learning has also been studies in “Su, J.C., Maji, S. and Hariharan, B., 2019. Boosting Supervision with Self-Supervision for Few-shot Learning. arXiv preprint arXiv:1906.07079.” and “Gidaris, S., Bursuc, A., Komodakis, N., Pérez, P. and Cord, M., 2019. Boosting Few-Shot Visual Learning with Self-Supervision. arXiv preprint arXiv:1906.05186.” This paper uses a self-supervised method called “pseudo labeling”, but other self-supervised methods should also be compared.

----------------------

Update after rebuttal:

Thanks for the new experiments on miniImageNet! However, I still have concerns about the novelty of the proposed idea (clustering+few-shot learning) and its similarity to previous works. In addition, the new results on miniImageNet is not very encouraging (it is necessary to provide a fair comparison). Hence, I will keep my rating unchanged.


**Experience Assessment:**

I have published one or two papers in this area.

**Review Assessment: Checking Correctness Of Derivations And Theory:**

I carefully checked the derivations and theory.

**Review Assessment: Checking Correctness Of Experiments:**

I carefully checked the experiments.

**Review Assessment: Thoroughness In Paper Reading:**

I read the paper thoroughly.

---

> ### Author Response · Authors · 2019-11-15
> **Response to Review #1**
>
> Thanks for your review and for giving some constructive comments. Below are our replies to your concerns.
>
> 1) We do not agree that unsupervised few-shot learning is not a big issue in real-world applications. An implicit assumption in the current few-shot learning or meta-learning models (e.g., ProtoNets, MAML, RelationNet, LEO) is that the training and testing tasks are constructed from the same dataset. Although they are from different classes, the data distribution is similar (at the very least they are all natural images in ImageNet). We cannot expect that a few-shot learner well-trained on ImageNet will generalize well on Omniglot or Market1501. If it does work, it must involve extra transfer learning skills. On the other hand, in many real-world applications, we do face the problem of insufficient labelled data (such as biological data and medical data) for constructing learning episodes. For example, we have a large number of unlabeled astronomical images (as they are very expensive to label) and face the task of building a few-shot learner. Training a few-shot learner on ImageNet does not work, as the model would over-fit on the natural image distribution and would not generalize to astronomical data. Developing unsupervised few-shot learning is relevant and crucial for real-world applications. In our study, we use Market1501 (an example of very hard labelling datasets) to demonstrate the feasibility of our UFLST model (see Sec.4.5 in the paper).
>
> 2) Our model is quite different from the CACTUs by Hsu et al., although they also use pseudo labels to train a feature embedder by using off-the-shelf unsupervised feature learning methods, e.g., ACAI, BigGAN, InfoGAN, DeepCluster, and then carry out episodic learning by generating pseudo labels with the well-trained features. However, in their model, feature learning and few-shot learning are performed separately (note the objective of the feature embedder learning is not the objective of few-shot learning in their work). This implies that the clusters generated from the learned features are not necessarily suitable for constructing the episodic tasks. As a result, when testing on Omniglot, they only obtained 68.84% under the 5-way 1-shot learning scenario. It is worth noting that the initial result (via the well-learned embedder) already has an accuracy of 61.08%, indicating that episodic learning in their model has only a small contribution. On the other hand, our model has a much better performance of 97.03% on Ominiglot. The superior performance of our model is due to the iterative integration of two learning processes: progressive clustering and episodic learning. These two processes promote each other to achieve good performances.
> The other two works you mentioned (Gidaris et al. and Su et al.) aim to boost the performance of few-shot learning by using an auxiliary self-supervised loss function and their model training is completely supervised, which is very different from our goal of learning a few-shot learner using unlabeled data.
>
> 3) We have added the results on the Mini-ImageNet dataset, please see the "General Response". For a detailed comparison of our model to other unsupervised few-shot learning methods, please see Appendix G.
>
> 4) We have included the comparison to the baseline on Market1501 (see Table 2 in our paper). Also, in Table 1, we have compared our model to all unsupervised few-shot learning models we could find.

---

### Official Review · AnonReviewer3 · 2019-10-23
**Official Blind Review #3**

**Rating:** 6

**Review:**

In this paper, the authors have investigated the unsupervised few shot learning problem. They have proposed a method to learn an unsupervised few-shot learner via self-supervised training (UFLST), which consists of two alternate processes, progressive clustering and episodic training. Experimental results on several benchmark data sets validate the effectiveness. Overall, I think it is an interesting scenario and have the following comments.
(1) The first one the about the method. The authors have tried to describe their methods in a concrete way. Nevertheless, it is better to give some justifications from the theoretical aspect. It is vital for readers to convince the superiority.
(2) The second one is about the description. I suggest the authors to express as accurately as possible. For example, in the abstract, the authors state that ‘However, current few-shot learners are mostly supervised and rely heavily on a large amount of labeled examples.’ I do not agree with this statement since in few shot learning, there are only limited exemplars. There is a conflict between them and it will confuse the readers.

**Experience Assessment:**

I have published one or two papers in this area.

**Review Assessment: Checking Correctness Of Derivations And Theory:**

N/A

**Review Assessment: Checking Correctness Of Experiments:**

N/A

**Review Assessment: Thoroughness In Paper Reading:**

N/A

---

> ### Author Response · Authors · 2019-11-14
> **The convergence of our UFLST model is similar to the EM-style learning algorithm.**
>
> Thanks for the encouraging comments, and we would like to address your concerns as below.
>
> 1) The good performance of our model comes from two strategies, progressive clustering and episodic learning, and they are performed iteratively. This has the same spirit as the EM-style algorithm. Initially, we randomly guess the pseudo-labels by clustering data in the feature space. (In reality it is not completely random. As observed in [1], the performance of a randomly initialized convnet is above chance level. For example, a multilayer perceptron classifier on top of the last convolutional layer of a random AlexNet achieves 12% accuracy on ImageNet while the chance level is 0.1%. This implies that this weak signal can be exploited to bootstrap the discriminative power of our model.) Afterwards, we construct episodic tasks using the pseudo-labels to learn feature representations, which are improved compared to the previous ones. In the next round, these improved feature representations in return further improve the clustering quality. Overall, in our model, the progressive clustering and episodic learning facilitate each other to generate good performance. Dempster et al. [2] theoretically analyzed the convergence property of the EM-style (non-parametric) learning algorithm. Since we used a 4-layer convnet as the feature extractor, the theoretical analysis is very difficult. To our best knowledge, there is no theoretical work on the progressive learning of deepnets, although empirical studies have confirmed its convergence. Please refer to [3] and [4].
>
> 2) In the literature, the existing episodic learning-based methods for few-shot learning always assume that a large number of labeled data X is available for constructing episodes. We agree that the number of labeled data required is not as large as ImageNet, but it is still a big number, and this is infeasible in many real-world applications. Hence, we study unsupervised episodic learning for few-shot learning.
>
> References:
> [1] Noroozi, M., & Favaro, P. (2016, October). Unsupervised learning of visual representations by solving jigsaw puzzles. In European Conference on Computer Vision (pp. 69-84). Springer, Cham.
> [2] Dempster, A. P., Laird, N. M., & Rubin, D. B. (1977). Maximum likelihood from incomplete data via the EM algorithm. Journal of the Royal Statistical Society: Series B (Methodological), 39(1), 1-22.
> [3] Song, L., Wang, C., Zhang, L., Du, B., Zhang, Q., Huang, C., & Wang, X. (2018). Unsupervised domain adaptive re-identification: Theory and practice. arXiv preprint arXiv:1807.11334.
> [4] Ben-David, S., Blitzer, J., Crammer, K., Kulesza, A., Pereira, F., & Vaughan, J. W. (2010). A theory of learning from different domains. Machine learning, 79(1-2), 151-175

---

### Author Response · Authors · 2019-11-15
**General Response**

We thank all reviewers for their constructive comments and suggestions for improving the strength and clarity of this paper. Here we address some common concerns raised by the reviewers.  We have updated the text of the paper to further incorporate these suggestions and fix the typos.

1) The main contribution of our model is the iterative integration of unsupervised clustering and episodic learning. More specifically, in our model, the feature learning is guided by episodic training on the generated pseudo labels, and the learned features (which are improved compared to the previous ones) are then used to generate the next round clusters. The two steps are carried out in an iterative manner. Our experiment results show that features learned this way are much more suitable for few-shot classification under the unsupervised setting.

2) We have added the results onto the Mini-ImageNet dataset. The main results are showed below (under the 5-way 1-shot learning scenario):
***********************************************************
training from scratch                                                                                $25.17\%$
UMTRA without data Augmentation                                                      $26.49\%$
UMTRA+Shift+random flip +randomly change to grayscale              $32.80\%$
AAL-MAML+++ CHV                                                                                    $33.06\%$
AAL-ProtoNets+ CHV                                                                                 $37.67\%$
BiGAN CACTUs ProtoNets                                                                        $36.62\%$
DeepCluster CACTUs ProtoNets                                                              $39.18\%$
---------------------------------------------------------------------------------------------------------
UFLST  without data Augmentation                                                        $33.77\%$
***********************************************************
CHV means Random Crops (C), Random Horizontal Flips (H),  and Random Vertical Flips (V) respectively. For the detailed comparison, please see Appendix G.

Here we point out three key aspects:
(1) The in-class variation of the Mini-ImageNet is very large, which is hard for our 4-layer convnet model to capture the semantic meanings of images. Thus, clustering quality is not good enough and constructed episodic tasks are highly noisy. Despite this, compared to the model trained from scratch (before iterative training), our UFLST model shows consistent improvement ( $33.77\%$  vs. $25.17\%$).
(2) Direct comparison with other unsupervised few-shot learning models is quite unsuitable and unfair. For example, both the UMTRA and the AAL model use fancy data augmentation strategies to improve their performances, whilst we implement our model without any data augmentation. The CACTUs used the AlexNet or the VGG16 to first learn a very good feature embedder for the downstream feature clustering processes, whilst our model is only composed of a 4-layer convnet. Our purpose is more about demonstrating the learnability of the model on the task.
(3) We point out that the market1501 dataset that we carry out experiments on is more meaningful than the Mini-ImageNet dataset, because it is more suitable for the unsupervised setting and brings our model closer to real-world application.

---

### Decision · Program_Chairs · 2019-12-19

**Decision:**

Reject

**Comment:**

This paper proposes an approach for unsupervised meta-learning for few-shot learning that iteratively combines clustering and episodic learning. The approach is interesting, and the topic is of interest to the ICLR community. Further, it is nice to see experiments on a more real world setting with the Market1501 dataset.
However, the paper lacks any meaningful comparison to prior works on unsupervised meta-learning. While it is accurate that the architecture used and/or assumptions used in this paper are somewhat different from those in prior works, it's important to find a way to compare to at least one of these prior methods in a meaningful way (e.g. by setting up a controlled comparison by running these prior methods in the experimental set-up considered in this work). Without such as comparison, it's impossible to judge the significance of this work in the context of prior papers.
The paper isn't ready for publication at ICLR.